# Knockout of IRF7 Highlights its Modulator Function of Host Response Against Avian Influenza Virus and the Involvement of MAPK and TOR Signaling Pathways in Chicken

**DOI:** 10.3390/genes11040385

**Published:** 2020-04-02

**Authors:** Tae Hyun Kim, Colin Kern, Huaijun Zhou

**Affiliations:** 1Department of Animal Science, University of California, Davis, CA 95616, USA; taekim@ucdavis.edu (T.H.K.); ckern@ucdavis.edu (C.K.); 2Integrative Genetics and Genomics Graduate Group, University of California, Davis, CA 95616, USA

**Keywords:** avian influenza virus, chicken, CRISPR/Cas9, interferon, IRF7, knockout, RNA-seq

## Abstract

Interferon regulatory factor 7 (IRF7) is known as the master transcription factor of the type I interferon response in mammalian species along with IRF3. Yet birds only have IRF7, while they are missing IRF3, with a smaller repertoire of immune-related genes, which leads to a distinctive immune response in chickens compared to in mammals. In order to understand the functional role of IRF7 in the regulation of the antiviral response against avian influenza virus in chickens, we generated *IRF7^-/-^* chicken embryonic fibroblast (DF-1) cell lines and respective controls (*IRF7^wt^*) by utilizing the CRISPR/Cas9 (clustered regularly interspaced short palindromic repeats/CRISPR-associated protein 9) system. IRF7 knockout resulted in increased viral titers of low pathogenic avian influenza viruses. Further RNA-sequencing performed on H6N2-infected *IRF7^-/-^* and *IRF7^wt^* cell lines revealed that the deletion of IRF7 resulted in the significant down-regulation of antiviral effectors and the differential expression of genes in the MAPK (mitogen-activated protein kinase) and mTOR (mechanistic target of rapamycin) signaling pathways. Dynamic gene expression profiling of the host response between the wildtype and IRF7 knockout revealed potential signaling pathways involving *AP1* (activator protein 1), *NF-κB* (nuclear factor kappa B) and inflammatory cytokines that may complement chicken IRF7. Our findings in this study provide novel insights that have not been reported previously, and lay a solid foundation for enhancing our understanding of the host antiviral response against the avian influenza virus in chickens.

## 1. Introduction

The avian influenza virus (AIV) has significant impact on the poultry industry worldwide [1,2]. For instance, the highly pathogenic avian influenza (HPAI) outbreaks in the U.S. between 2014 and 2015 resulted in almost twelve and eight percent loss of layer chickens and meat turkeys, respectively (more than 50 million birds in total) [3,4]. In addition, AIVs are zoonotic and have posed great challenges to global public health [5,6]. However, current passive prevention practices such as quarantining and culling have been mainly applied to control AIV, partially due to our limited understanding of the antiviral response in birds compared to in mammals [7]. Improving our understanding of the avian host antiviral response could help to develop better strategies for controlling AIV in chickens and limiting zoonotic transmission, possibly by identifying potential biomarkers and therapeutic targets against AIV infection [8,9]. 

The coordination of Interferon Regulatory Factor 7 (IRF7) and IRF3 on the initiation of the type I interferon (IFN) response to establish the antiviral state is well characterized in mammals; however, chickens may utilize a different modulation system due to a smaller repertoire of immune-related genes in avian species [9,10]. Particularly, chickens are missing IRF3, and only have IRF7, which has less than 40% of amino acid sequence identity to its mammalian orthologues [11,12,13]. Therefore, studying the chicken IRF7 could be a great starting point to elucidate the underlying cellular mechanisms of host immune regulation against AIV in chickens. 

In our previous study using stable IRF7 overexpression and knockdown in chicken DF-1 cell lines with double-stranded RNA (dsRNA) analog poly(I:C) (polyinosinic-polycytidylic acid) inductions, we demonstrated the conserved function of IRF7 as a type I IFN modulator [14]. Another study also suggested a similar role of IRF7 by the siRNA knockdown of IRF7 in chicken embryonic fibroblasts (CEFs), which limited *IFNA*, *IFNB* and *STAT1* (signal transducer and activator of transcription 1) expression and increased Newcastle disease virus replication [15]. While these knockdown approaches such as RNA interference (RNAi) have successfully generated partial IRF7 loss-of-function phenotypes in chickens, and generated some novel insights into the role of IRF7 in the regulation of the host response to virus infection in poultry, RNAi has its own technical limitation of incompleteness of knockouts [16]. In this study, we decided to apply a CRISPR/Cas9 (clustered regularly interspaced short palindromic repeats/CRISPR-associated protein 9) approach to completely knockout (KO) the chicken IRF7 in the DF-1 cell line, in order to elucidate, in more depth, the underlying molecular mechanism of the modulation of the host response by IRF7 in chickens.

To understand the functional role of chicken IRF7 on the regulation of the host response to AIV at the genome-wide level, we applied a KO assay followed by a functional genomics approach (RNA-seq). First, the complete KO of IRF7 was achieved by the mutation chain reaction (MCR) which utilizes the CRISPR/Cas9 system to induce spontaneous homozygous mutation [17]. The viral replication phenotype was measured using two LPAIV (low pathogenic AIV) strains to examine the consequence of IRF7 deletion in vitro. Next, we applied RNA-seq to identify transcriptomic landscape differences behind the phenotype resulting from IRF7 KO and further reveal genes and signaling pathways associated with the regulatory function of chicken IRF7 in the context of AIV infection.

## 2. Materials and Methods 

### 2.1. Construction of IRF7 Knockout MCR Plasmids

We constructed MCR plasmids targeting chicken IRF7 by modifying the pSpCas9(BB)-2A-Puro (px459, a gift from Feng Zhang-Addgene plasmid 62988) [18]. Three guide RNAs (gRNAs) targeting a region in exon 2 of IRF7 were designed, and homology arms (~0.5 kb each), in which the two genomic sequences are immediately adjacent to either side of the target double strand break site, were amplified using specific primers: HA1 forward 5′-cca gag agc agg gag ggg aa-3′, HA1 reverse 5′-gtc cgg gtc gat cca gca ga-3′, HA2 forward 5′-cgc cgt atc ttc cgc atc cc-3′, HA2 reverse 5′-agc ctc cct tga agc ccc tg-3′—and assembled with PciI/NotI double digested px459 by Gibson assembly [19]. Each gRNA was cloned into the HAs assembled px459 and three different MCR-IRF7-Exon2 vectors were generated (Figure 1A).

### 2.2. IRF7 Knockout Cell Clone Generation

Immortalized CEF, DF-1 cells (ATCC, Manassas, VA, USA) were cultured in Dulbecco’s Modified Eagle’s medium (Thermo Fisher Scientific, Waltham, MA, USA) supplemented with 10% fetal bovine serum (Thermo Fisher Scientific) and 1x Antibiotic-antimycotic (Thermo Fisher Scientific), and incubated at 37 °C in a humidified atmosphere containing 5% CO_2_. Each MCR-IRF7-Exon2 vector was transfected into DF-1 cells using the Lipofectamine 3000 reagent (Thermo Fisher Scientific) according to the manufacturer’s protocol. Approximately 5 × 10^3^ cells were seeded into a 150 mm culture dish (Thermo Fisher Scientific) 48 h after transfection with puromycin (3 μg/mL) added to the 20 mL culture media. We selected 12 cell clones for each gRNA target by Pyrex® cloning cylinders (Corning, Corning, NY, USA) according to the manufacturer’s protocol after two weeks of transfection. Each clone was sub-cultured into a 48-well plate where 10% of cells were used for genotyping, and 90% of cells were cryopreserved, when each well reached confluency for subsequent analyses.

### 2.3. IRF7 Knockout Clone Genotyping

For genomic DNA extraction, we used 50 µL of the QuickExtract DNA extraction solution (Epicentre, San Diego, CA, USA) for each clone according to the manufacturer’s protocol. PCR analysis was performed to detect junctional bands corresponding to MCR insertion into the IRF7 exon2 locus, as well as the absence of a PCR band derived from the wild-type locus (Figure 1A, Appendix A).

### 2.4. Cas9-Puro Cell Line Generation

To generate the piggyBac-Cas9-Puro plasmid (Figure 1A), the SpCas9-Puro cassette was cloned in from px459 into the modified PB513B plasmid backbone (System Biosciences, Mountain View, CA, USA). The PiggyBac transposase plasmid was co-transfected with the piggyBac-Cas9-Puro into DF-1 cells using the Lipofectamine 3000 reagent (Thermo Fisher Scientific) to induce efficient and stable integration. Puromycin (3 μg/mL) was added to the culture media 48 h after transfection, and stably integrated cell lines were selected for two weeks. Cas9 activity was examined by multiplex gRNA transfection followed by genomic DNA PCR. Two gRNAs targeting the exon 2 and exon 10 loci of IRF7 were co-transfected into the Cas9-Puro cell line, and the deletions of the genomic DNA region between gRNA targets were examined by genomic DNA PCR (Appendix A). 

### 2.5. Quantitative Reverse Transcriptase PCR

Total RNA was isolated from approximately 1 million cells using the Direct-zol RNA MiniPrep Kit (Zymo research, Irvine, CA), and complement DNA (cDNA) was synthesized from total RNA (500 ng) using Verso cDNA Synthesis Kit (Thermo Fisher Scientific). Quantitative reverse transcriptase PCR (qRT-PCR) was performed using the QuantStudio 3 Real-Time PCR system (Thermo Fisher Scientific) with the SYBR Select Master Mix (Life Technologies). *IRF7*, *IFNA* and *IFNB* expressions were normalized to that of the *GAPDH* (glyceraldehyde 3-phosphaste dehydrogenase) gene using the ΔΔC_T_ method (Appendix A) [14,20].

### 2.6. AIV and In Vitro Infection

A/Chicken/California/2000 (H6N2) and A/Chicken/California/1999 (H10N7) low pathogenic avian influenza virus (LPAIV) strains were kindly provided by Dr. Rodrigo Gallardo (University of California, Davis) and Dr. Peter Woolcock (University of California, Davis, California Animal Health & Food Safety (CAHFS)), respectively. Each LPAIV was propagated in Madin-Darby Canine Kidney (MDCK) cells as described previously [21]. All in vitro AIV infections were performed using CellBIND 12 well tissue culture plates (Corning, Corning, NY, USA) with 1 × 10^6^ cells per well at the seeding. For the viral replication kinetics phenotype, established DF-1 cell lines were infected with either H6N2 or H10N7 at a multiplicity of infection (MOI) of 0.01 with 0.05 μg/mL TPCK-trypsin in DMEM. Culture supernatants were collected at 0, 12, 24 and 36 hours post-infection (hpi) and the viral titer of each sample was measured by the plaque assay using MDCK cells [22]. For the transcriptome profiling study, cell lines were infected with either mock or H6N2 at an MOI of 1 with 0.05 μg/mL TPCK-trypsin in DMEM. Trizol reagent (Thermo Fisher Scientific) was directly added to the cell monolayer, after it was washed twice with PBS, to extract the total RNA at 0, 2, 6 and 12 hpi.

### 2.7. RNA Sequencing and Data Analysis

A total of 24 cDNA libraries were prepared from three biological replicates of each time point (0 h (mock), 2-, 6- and 12 hpi *IRF7^wt^* or *IRF7^-/-^*). Directional RNA sequencing libraries were prepared from poly-adenylated RNA and sequenced with the Illumina HiSeq4000, which generated over 20 million 100 bp paired-end reads per sample, except for 2 hpi samples where 150 bp paired-end reads were generated. The read files from RNA-seq analysis have been deposited in NCBI’s Gene Expression Omnibus with accession number GSE119590. We checked the quality of each library using fastQC (Babraham Institute, Cambridge, UK, version 0.11.6) and trimmed the adaptor sequence with TrimGalore (Babraham Institute, Cambridge, UK, version 0.4.5). We aligned the trimmed fastq files to the galGal5 chicken genome using the STAR aligner with NCBI annotation release 103 [23]. Unmapped reads were aligned against the H6N2 (A/chicken/CA/6643/2001) genome [24]. Raw read counts were extracted by HTSeq from each aligned bam file and used to identify differentially expressed genes (DEGs) [25]. Both DESeq2 and EdgeR R packages were used to identify DEGs and the DEG sets from both packages were combined (false discovery rate (FDR) < 5% or 1% in any one of the packages) [26,27,28]. The combined DEG lists were further filtered by removing the low expression genes if both samples of a given comparison had fragments per kilobase per million reads (FPKM) value less than 1. Functional annotations for significantly differentially expressed genes were performed using DAVID 6.8 [29,30]. The enriched gene ontology (GO) terms on biological processes and the pathways obtained from DAVID functional analysis were filtered for significance by gene count ≥5 and *p*-value < 0.05. Additional pathway analyses were performed using the Ingenuity Pathway Analysis (IPA) software (QIAGEN, Germantown, MD, USA). RNA expression of the MCR target region was visualized by the Integrative Genomics Viewer (IGV, version 2.4.13) [31]. Heat maps and k-means clustering analysis diagrams were analyzed using in-house R and Python scripts. We applied the gap statistics and sum of squared error methods to evaluate the optimum number of clusters [32].

## 3. Results

### 3.1. Establishment of IRF7 Knockout DF-1 Cell Clones

We applied the MCR method, which uses homology directed repair (HDR), to disrupt chicken IRF7 in a highly efficient manner. Three gRNAs targeting within a 6bp region of IRF7 exon 2 were designed to address the potential off-target effects induced by the CRISPR/Cas9 system, since the MCR insert constitutively expresses both SpCas9 and gRNA (Figure 1A) [33]. We were able to use an identical set of homology arms (HAs) flanking the gRNA-SpCas9-Puro transgenes for all the gRNAs, as the target sites of the gRNAs were adjacent to each other (Figure 1A). PCR analysis of the IRF7 exon 2 locus in individual cell clones confirmed the precise gRNA- and HDR-directed genomic insertion of the MCR construct in the IRF7 exon 2 locus (Figure 1B). By the MCR and puromycin selection, we were able to achieve 50% of bi-allele KO cell clones, which only required genotyping less than 10 clones per gRNA. The genotypes we observed were either bi-allele genomic insertions or single-allele insertions. We randomly selected one clone for each gRNA target, which served as biological replicates for *IRF7^-/-^* (Figure 1B, marked asterisk). We also confirmed disrupted *IRF7* transcription and interrupted mRNA splicing in the *IRF7^-/-^* cells at the MCR region from RNA-seq analysis (Figure 1C).

Since the *IRF7^-/-^* cells have a targeted insertion of SpCas9 and puromycin resistance genes in their genome, we generated corresponding control *IRF7^wt^* cells that have SpCas9 and puromycin resistant genes randomly integrated by piggyBac transposition (Figure 1A, right panel) [34]. We assessed the activity of SpCas9 and puromycin resistance genes in the *IRF7^wt^* cells by co-transfecting two gRNAs and selecting with puromycin, respectively. SpCas9 activity was confirmed by successfully detecting the 2.7kb-deletion alleles as a result of gRNA multiplexing (Appendix A).

### 3.2. LPAIV In Vitro Infection

We next sought to determine whether IRF7 deficiency affects the viral replication phenotype in vitro. *IRF7^-/-^* and *IRF7^wt^* cells were infected with H6N2 at an MOI of 0.01 and then supernatant was collected at 0, 12, 24 and 36 hpi to measure viral titers (Figure 1D). We observed significantly an increased H6N2 viral replication level in the *IRF7^-/-^* cells compared with in the *IRF7^wt^* cells at 12 hpi. Another strain of LPAIV H10N7 showed a higher viral replication level as a result of IRF7 knockout (Appendix A). Transcriptome analysis corresponds to this pattern where we observed a relatively lower fraction of host mRNA and a higher fraction of viral RNA in *IRF7^-/-^* cells than in *IRF7^wt^* cells at 6 and 12 hpi, yet not at a significant level (Appendix A).

### 3.3. The Effect of IRF7 Knockout on the Host Response Against H6N2 Infection 

To elucidate the molecular mechanisms of chicken IRF7 in the context of AIV infection in depth, we directly contrasted the transcriptomes between the *IRF7^wt^* and *IRF7^-/-^* cells at each time point. Differential gene-expression analysis indicated that 180 genes were significantly upregulated, and 164 genes were significantly downregulated in *IRF7^-/-^* cells compared to their expressions in *IRF7^wt^* cells without infection (FDR < 5%; Figure 2A and Appendix A). Upon infection, there were 138 down- and 100 up-regulated DEGs at 2 hpi due to IRF7 deletion, and the number of DEGs increased to 441 down- and 483 up-regulated genes at 6 hpi. At 12 hpi, fewer genes were differentially expressed (366 up- and 300 down-regulated) compared to at 6 hpi (Appendix A). Of note, the numbers of DEGs observed comparing *IRF7^wt^* with *IRF7^-/-^* cells were similar between up-regulated genes and down-regulated genes at all time points. We used these DEGs to perform functional enrichment analyses by DAVID to identify enriched biological process gene ontology (GO) functions and pathways that were regulated by IRF7 (Figure 2B). Upon infection at 2 hpi, we observed enriched immune response related terms such as antigen processing and presentation, as well as the regulation of the inflammatory response. The mTOR (mechanistic target of rapamycin) signaling GO term and its downstream process actin polymerization were also enriched at 6 hpi. Protein processing associated terms were significantly enriched across all time points (Figure 2B). Heatmaps with fold changes of genes in the influenza A pathway and significantly enriched pathways across all time points using all individual DEGs are presented in Figure 2C–E. *IL18* (interleukin 18), *MAPK13* (mitogen-activated protein kinase 13; *p38*) and *CHUK* (conserved helix-loop-helix ubiquitous kinase; *IKKA*) were up-regulated in influenza A and the MAPK signaling pathways, whereas MHC (major histocompatibility complex) protein family genes (*BF1*, *BF2*, *DMA*, *DMB2* and *BLB2*) displayed down-regulated expression across all time points once IRF7 was deleted. 

### 3.4. Comparative Analysis of DEGs Between IRF7 Knockout and Overexpression

Our previous work has identified candidate genes that were regulated by IRF7 in the context of H6N2 infection by applying an inducible overexpression-based gain-of-function approach [35]. We infected IRF7 overexpression cells (*CuO-IRF7*: 2-fold *IRF7* overexpression) and their respective controls with H6N2 (1 MOI) and performed RNA-seq at 0 h (mock) and 6 hpi. To identify candidate genes regulated by IRF7, we directly compared the datasets of IRF7 KO and overexpression that were contrasted to their respective controls at 0h and 6 hpi (*IRF7^-/-^*/*IRF7^wt^* vs. *CuO-IRF7*/*Control*; FDR < 5%, Figure 3). A total of 245 common DEGs between the two datasets were identified, which were grouped into four distinct clusters: same direction of regulations (upregulated or downregulated) or opposite directions of the regulation between knockout and overexpression. For example, cell adhesion molecules (CAMs) such as laminins, integrins and collagens had up-regulated expression if IRF7 expression was altered in any direction. On the other hand, genes enriched for the cellular response to laminar fluid shear stress or small GTPase-mediated signal transduction were down-regulated for both *IRF7* KO and overexpression. Of particular interest, opposite directions of regulation were observed. TOR-signaling and I-kappaB phosphorylation-associated DEGs were up-regulated due to IRF7 deletion and down-regulated once *IRF7* was overexpressed. In addition, the expressions of antigen processing and presentation-related genes and immune response genes were positively correlated with *IRF7* expression. Interestingly, the MHC I family gene *BF2* was down-regulated in both IRF7 KO and overexpression, whereas the expression of MHC II family genes (*DMA*, *DMB2*, *BLB2*) corresponded with the IRF7 expression level.

### 3.5. Differential Expression Analysis of the Host Response upon H6N2 Infection

Next, we compared the host responses between the cell lines upon H6N2 infection, by comparing infected cells to the uninfected cells at each time point of infection, and identified which genes and pathways were potentially modulated by IRF7 in the host response (Figure 4A, Appendix A). To further analyze which biological process was possibly suppressed due to IRF7 deletion, or alternatively activated in the antiviral response to compensate for the absence of IRF7, we first compared the list of DEGs between cell lines at each time point (Figure 4B). Next, we took the DEGs that were specific to each cell line and performed GO enrichment analysis at each time point (Figure 4C). Enriched terms demonstrated IRF7 to be involved in a wide range of biological processes including mitosis, cell cycle, protein modification, transcription and translation. At 6 hpi, genes involved in NF-κB (nuclear factor-κB) signaling regulation were enriched only in the IRF7 KO cells. However, we were not able to specify which functions are mainly regulated by IRF7 with this analysis, mainly because the expression levels of individual genes did not show significant differences between the cell lines, despite the differences in the DEG lists and functional annotations. 

### 3.6. Dynamic Gene Expression Profile Analysis Revealed an Alternative Pathway to IRF7 Regulation

To evaluate the global gene expression profile alteration due to IRF7 KO in the host response to H6N2 infection, we performed *k*-means clustering among the 5091 DEGs that were obtained from previous analyses in Figure 2 and Figure 4. A total of six clusters were identified based on their expression patterns (Figure 5A, Appendix A). A predominant expression pattern shift was observed from Cluster V to Cluster II due to the deletion of IRF7, which was almost 60% (630 out of 1067, Figure 5B, Appendix A) of the genes from Cluster V. Gene ontology based on the *IRF7^wt^* DEGs of each cluster revealed that the genes in Cluster V have enriched functions associated with translation, inflammatory response and immune response (Figure 5C). These 630 shifted genes were similarly enriched with GO terms such as translation, immune response and defense response to pathogens (Appendix A). Moreover, the expression dynamics of the genes changed from the stable-up-down pattern to constant up pattern once IRF7 was deleted. Based on this observation, we hypothesized that the deletion of IRF7 resulted in an insufficient host response against AIV, hence the expression of alternative inflammatory response and immune response genes might be needed to compensate for the effect of deleted IRF7. This hypothesis was demonstrated in the influenza A pathway (Figure 5D, Appendix A) in which seven out of twelve genes from Cluster V in the *IRF7^wt^* cells were shifted to Cluster II in *IRF7^-/-^* cells. These genes include key immune and inflammatory response genes, such as inflammatory cytokines (*IL12* (interleukin 12) and transcription factors (*FOS* (Fos proto-oncogene; AP1 transcription factor subunit) and *RELA* (RELA proto-oncogene; NF-κB subunit, p65)), as well as their upstream molecules (*IKBKE* (inhibitor of kappa light polypeptide gene enhancer in B-cells, kinase epsilon) and *NFKBIB* (NF-κB inhibitor beta)).

## 4. Discussion

IRF7 was identified as the regulator of the type I IFN response in an *Irf7^-/-^* mouse model, in which the *Irf7^-/-^* mouse was significantly more vulnerable to viral infections, with markedly inhibited type I IFN expression, than the wild type [10,36]. The significant role of IRF7 in the regulation of the host anti-viral response was further elucidated in humans, in which the IRF7-deficient patient had impaired type I and III IFN responses against primary infection and experienced a life-threatening influenza infection [37]. 

Interaction of highly homologous IRF3 and IRF7 (oligomerization) is recognized as a key regulatory mechanism of the transcriptional activation of type I IFNs in initiating the host antiviral state in mammals [38]. Chicken IRF7 was first identified and named as IRF3 due to its high degree of similarity to mammalian IRF3, but was renamed as IRF7 when subsequent studies revealed that IRF3 is absent in avian species [39,40]. We took a de novo transcript assembly approach to identify potential chicken IRF3 at the transcript level, in order to eliminate potential annotation issues in the chicken reference assembly, yet we did not detect the chicken IRF3 [14]. Based on the function of IRF7 homologs in mammals and missing a member of the IRF3/IRF7 dyad, we hypothesized that chicken IRF7 is a central transcription factor in the antiviral response against AIV infection in chickens.

In this study, we employed a functional genomic approach to elucidate the functional role of chicken IRF7 in the host response against AIV infection in chickens. Our work has first revealed that one of type I IFN, *IFNB*, but not *IFNA,* was selectively responsive to IRF7 KO in chicken DF-1 cells. The type I IFN induction models from in vitro studies using mouse embryonic fibroblast or primary tracheal epithelial cells of *Irf7^-/-^* mice showed an abolished production of type I IFNs upon viral infection [10,41]. In our previous studies, we consistently observed a significant upregulation of *IFNB* expression as a result of IRF7 overexpression in the DF-1 cell lines upon dsRNA induction or H6N2 infection [14,35]. On the other hand, *IFNA* mRNA levels were not significantly affected by the level of IRF7 expression [14,35], and a similar pattern was observed in this study (Figure 1E–G). Although DF-1 cells are capable of mounting a type I IFN response, the level of response is possibly lower than that of primary CEFs due to a higher suppressor of cytokine signaling 1 (*SOCS1*) expression in DF-1 [42]. Therefore, additional studies in different cell types such as lymphocytes, macrophages or dendritic cells could further address the context and cell type specific type I IFN regulation by IRF7.

We also demonstrated that the deletion of IRF7 resulted in the higher replication of LPAIVs in vitro, which further suggests an anti-viral role of IRF7 in chickens. This phenotype was in agreement with the susceptible phenotype observed in *Irf7^-/-^* mice against viral infections (herpes simplex virus, encephalomyocarditis virus or influenza virus) [10,43]. Functional enrichment of DEGs contrasting between *IRF7^wt^* and *IRF7^-/-^* cells identified the MAPK pathway as one of the significant pathways altered due to IRF7 KO (Figure 2D). The MAPK pathway is an important signaling cascade that controls a wide range of cellular responses such as proliferation, differentiation, and the immune response [44]. Among the known four MAPK family members, p38 MAPK (*MAPK13*) was upregulated as a result of IRF7 KO. The inflammatory response due to activation of p38 by influenza virus infection has been reported, and the pro-viral function of p38 MAPK was confirmed in a mouse model where the inhibition of p38 MAPK significantly protected mice from lethal influenza infection (H5N1) [45]. In this perspective, the upregulation of the MAPK pathway in the *IRF7^-/-^* cells could possibly explain the increased viral replication phenotype of the IRF7 KO cells. Our observations were in line with the study comparing transcriptomes between the wildtype and *Irf3^-/-^Irf7^-/-^* double KO (DKO) mice infected with PR8 (H1N1), in which the upregulation of MAPK signaling pathway was observed to correlate with susceptible phenotypes of mutants such as significant weight loss, a higher mortality and a higher viral load with infection [43]. A validation of the chicken IRF7 KO susceptibility phenotype against AIV at a systemic level is warranted to elucidate the underlying molecular mechanism. 

The integration of both gain- and loss-of-function approaches employed in this study provided a unique opportunity to distinguish between the genes that were either modulated by IRF7 or altered as a result of IRF7 perturbation. For instance, we observed differential expression of genes that are involved in the membrane trafficking pathways, such as endocytosis, lysosomes and phagosomes, as a result of IRF7 KO after infection, as well as at baseline (Figure 2). Of particular note, both MHC I and MHC II family genes were downregulated due to IRF7 deletion. Further analysis comparing the DEGs of IRF7 KO and overexpression revealed that only MHC II family genes were positively correlated to IRF7 expression, while the expressions of MHC I genes were downregulated in both IRF7 KO and overexpression (Figure 3). This suggest an important role of chicken IRF7 in the regulation of antigen presentation against viral infection through the MHC II family of genes. Cell adhesion molecules, particularly integrins, play important roles as mediators in surveillance and the regulation of immune cell functions [46], and we observed the up-regulation of integrins as a result of altered IRF7 expressions in any direction, even in the absence of infection. This could imply that the IRF7-dependent host response in chicken is a finely-tuned cell process, and IRF7 may be an important regulator of cell state determination between the homeostatic state and antiviral microenvironments in the cells throughout the host-pathogen interaction.

In addition, a negative correlation between mTOR and IRF7 was corroborated: IRF7 KO resulted in the upregulation of *MTOR*, while the overexpression of IRF7 led to the downregulation of *MTOR* (Figure 3, Appendix A). The mTOR signaling pathway is highly conserved in eukaryotes from yeast to humans and plays a central regulator role in cellular activities such as cell proliferation, metabolism, survival and differentiation [47,48]. This may suggest chicken IRF7 directly regulates mTOR signaling to modulate cellular processes in the host response against AIV. This is consistent with the previous report that indicated the repressive translational regulation of *Irf7* mRNA by the mTOR signaling pathway [49]. The mTOR signaling pathway is triggered during the host innate immune response and reconfigures cellular metabolism to provide the cellular environment for a subsequent immune response in a cell-type-specific manner [47,50,51]. The mTOR signaling pathway coordinates metabolic processes through forming two distinct multi-protein complexes, mTOR complex 1 (mTORC1) and mTORC2 [47]. Our functional enrichment analyses of IRF7 KO revealed alterations of pathways usually mediated by mTORC1 such as the metabolic process, protein synthesis (translation and ribosome) and protein turnover (protein ubiquitination) within the host response (Figure 3 and Figure 5). Moreover, IRF7 deletion resulted in the upregulation of *RICTOR* (rapamycin insensitive companion of mTOR), a component of mTORC2, which functions as an effector of PI3K/Akt signaling and cytoskeleton rearrangement [47]. Accordingly, we observed an expression pattern shift of PI3K and upregulation of AKT3 in the *IRF7^-/-^* cells compared to in the *IRF7^wt^* (Appendix A). Our previous study provided another line of evidence supporting the mTOR-mediated IRF7 modulation of the host response, in which GO terms such as arginine and proline metabolism, pyrimidine metabolism and the regulation of the actin cytoskeleton were significantly enriched from the DEGs of IRF7 overexpression vs. wildtype [35]. 

Comparing the host responses between the *IRF7^wt^* and *IRF7^-/-^* cells enabled the uncovering of the pathways that work together with IRF7 in the host antiviral response against H6N2 infection. Significantly more genes (1200+) differentially expressed between the *IRF7^-/-^* cells and *IRF7^wt^* cells at 12 hpi (Figure 4A) highlighted the potential pathways that might compensate for the host response, due to the lack of IRF7, in response to AIV. Dynamic gene expression profiling applying *k*-means clustering further revealed a set of genes that had their expression pattern shifted due to IRF7 deletion with AIV infection (Figure 5). The shifts of immune response- and inflammatory response-related genes again emphasized a significant modulator role of IRF7. Particularly, the constant augmentation of transcription factor NF-κB (nuclear factor kappa B; subunit *RELA*, inhibitor *NFKBIB*) and AP1 (activator protein 1; subunit *FOS*) expression during infection due to IRF7 deletion suggests that these factors complement IRF7 in the host antiviral response by modulating inflammatory cytokines and type I IFNs in chickens, as in mammals (Figure 5D, Appendix A) [52]. This is in line with the observation from a *Irf3^-/-^Irf7^-/-^* (DKO) mice study in which the DKO mice displayed impaired interferon activation and an increased inflammatory response upon influenza A virus infection [43]. In addition, *Irf3^-/-^Irf7^-/-^* DKO mice as well as *Irf3^-/-^Irf5^-/-^Irf7^-/-^* triple KO mice infected with dengue virus had demonstrated enriched inflammatory process-related genes at 24 hpi compared to the wildtype animals [53]. 

A schematic model of early host response associated with chicken IRF7 is proposed in Figure 6, based on the current study and our previous in vitro studies [14,35]. The entry of AIV is recognized by pattern recognition receptors (PRRs) sensing viral moieties. TLR3 (toll-like receptor 3) and MDA5 (melanoma differentiation-associated protein 5, IFIH1) respond to dsRNA, and TLR7 responds to single-stranded RNA (ssRNA) during AIV infection in chickens [54,55,56]. This leads to a subsequent signaling cascade via adaptor molecules and protein kinases that activate the key transcription factors (AP1, NF-κB and IRF7). The MAPK and PI3K/Akt pathways are potentially engaged in this signaling module in addition to the classical type I IFN induction pathway activating IRF7 by IkB kinases (IKK) TBK1 (TANK binding kinase 1), IKKε (IKBKE) or IKKα (CHUK) [13,57,58]. Overall, our data argue that IRF7 is a major modulator of the host antiviral response, and that the transcription factors NF-κB and AP1 are also capable of regulating both IFNA and INFB in addition to the inflammatory cytokines. Among the wide range of biological processes in the antiviral responses that are modulated by IRF7, two modes of regulatory mechanism were speculated: a cascade of downstream type I IFN responses, and the direct modulation of genes by IRF7 binding to their regulatory regions (Figure 6). Genome-wide chromatin immuno-precipiation (ChIP)-seq approach using CRISPR-Cas9-mediated knock-in could further dissect the molecular basis of the IRF7-mediated antiviral response.

In summary, we integrated genome editing and RNA sequencing to reveal some novel findings on the role of chicken IRF7 in the context of AIV infection at the whole transcriptome level. This study suggests the conserved function of chicken IRF7 as a major modulator of the antiviral response, despite the considerable differences between the chicken and mammalian innate immune systems. Moreover, our results suggest a novel insight on the chicken type I IFN response, in which MAPK and mTOR signaling may play important roles in modulating the antiviral response against AIV. Lastly, we found evidence that implicates the involvement of other important mediators working together with IRF7 in the host antiviral response in chickens. Although further systemic-level studies using different cell types are warranted, our results have provided the first line of evidence that has enhanced our understanding of the role of IRF7 on the host antiviral response to AIV in chickens. 

## Figures and Tables

**Figure 1 genes-11-00385-f001:**
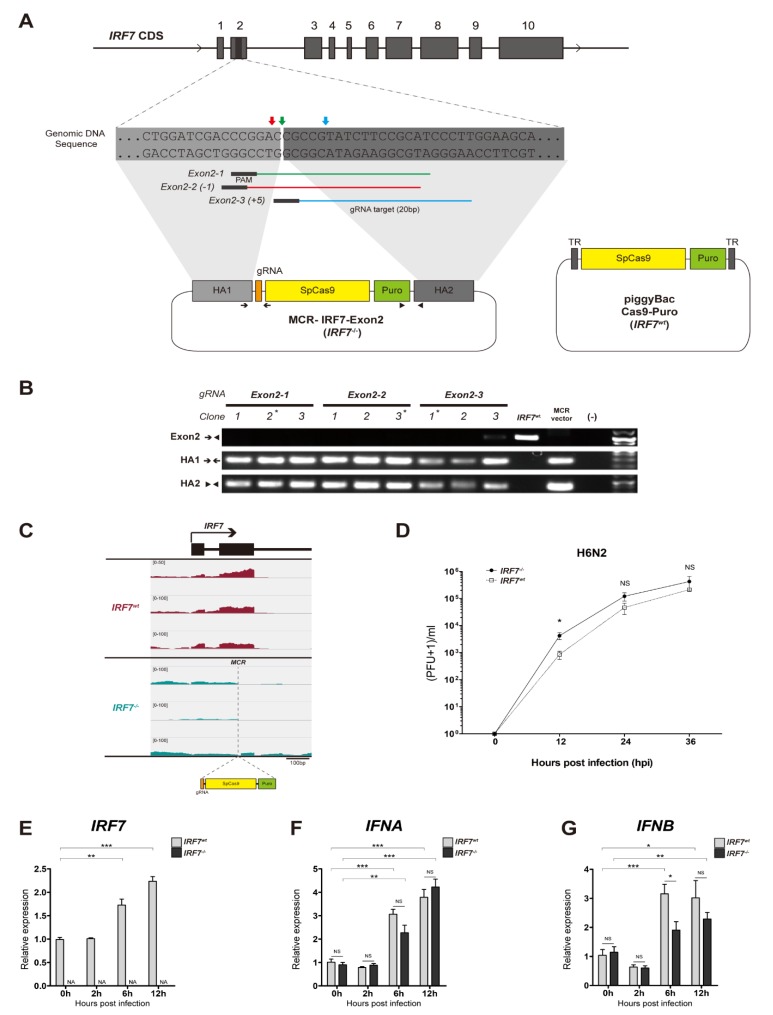
The establishment of *IRF7^-/-^* and control (*IRF7^wt^*) DF-1 cell lines. (**A**) Left panel: a scheme outlining the IRF7 knockout by the mutagenic chain reaction (MCR). An MCR plasmid targeting the second exon of chicken IRF7 was designed. Three adjacent guide RNAs (gRNAs) targeting the IRF7 exon 2 locus were selected, and each gRNA was cloned into the designed MCR plasmid. The locations of the PCR primers used for the analysis of the genomic insertion site are indicated. Right panel: the SpCas9-puromycin resistant gene construct from the MCR plasmid was cloned into the piggyBac transposon expression vector to generate control (*IRF7^wt^*) DF-1 cells. (**B**) PCR analysis of bi-allelic MCR insertion cell clones for each gRNA is shown. The absence of a PCR band derived from the IRF7 exon2 locus (Exon2) and the presence of junctional bands corresponding to MCR insertion into the chromosomal IRF7 exon 2 locus (HA1, HA2) are shown for MCR cell clones. (**C**) RNA-seq alignment visualization of *IRF7^wt^* (upper panel) and *IRF7^-/-^* (lower panel) in non-infected cells (0 h). The genomic region of exons 1 and 2 of IRF7 is shown. The gray vertical dashed line indicates a putative double strand break and homology directed repair site. (**D**) Culture supernatants were collected at each time point and the viral titer was determined by the plaque assay as (plaque forming unit (PFU)+1)/ml using Madin-Darby kidney (MDCK) cells. (**E**–**G**) The qRT-PCR analysis of *IRF7*, *IFNA* and *IFNB* transcripts in *IRF7^-/-^* and *IRF7^wt^* cells upon H6N2 infection (1 multiplicity of infection (MOI)). All data are shown as mean ± SEM from six biological replicates (* *p* < 0.05, ** *p* < 0.01, *** *p* < 0.001, NS: not significant; two tailed *T*-test).

**Figure 2 genes-11-00385-f002:**
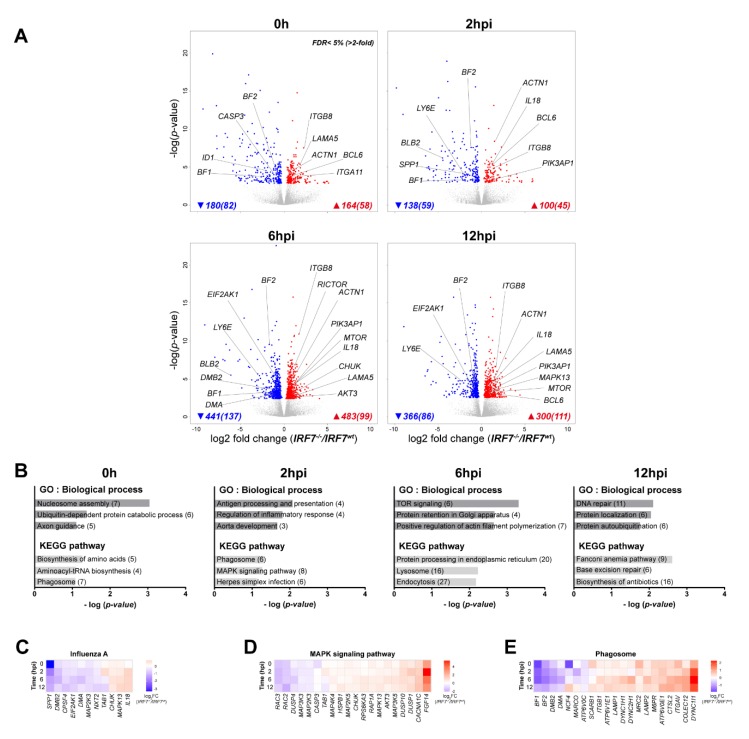
IRF7 deletion altered transcriptome regulation in DF-1 cells (**A**) Volcano plots of transcriptomic differences between *IRF7^-/-^* and *IRF7^wt^* cells at each time point; the colored dots correspond to significant differentially expressed genes (DEGs; False discovery rate (FDR) < 5 %). The numbers of DEGs are shown at the bottom of each plot, and the numbers of genes that have more than 2-fold changes are in the parentheses. (**B**) Gene ontology (GO) and pathway analysis by DAVID 6.8 using DEGs from each time point. The numbers of genes enriched in each biological process are in parentheses. (**C**–**E**) Heatmaps showing the expression fold change of DEGs in influenza A, MAPK (mitogen-activated protein kinase) signaling and the phagosome pathways at all time points (*IRF7^-/-^*/*IRF7^wt^*). Lists of DEGs from all time points in each pathway were combined to generate each heatmap.

**Figure 3 genes-11-00385-f003:**
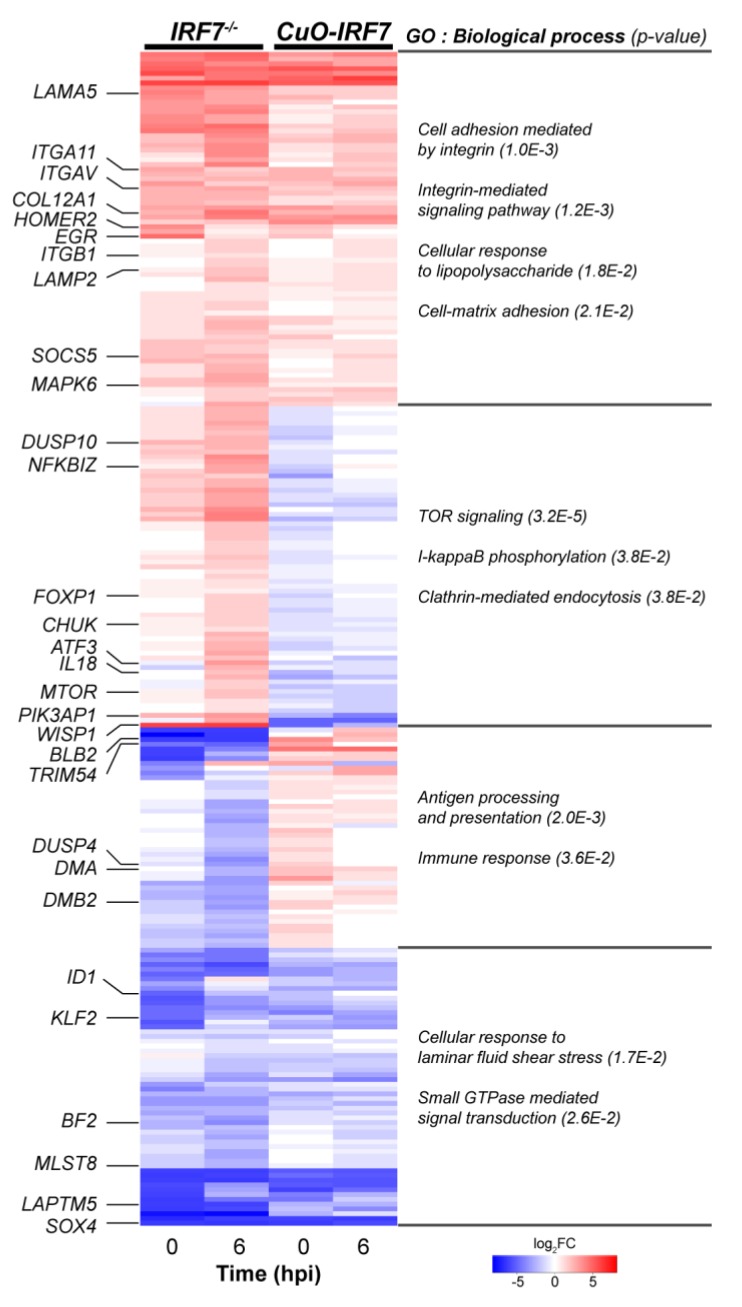
Comparative analysis of IRF7 knockout and overexpression. The expression profile of 245 common DEGs between IRF7 knockout (*IRF7^-/-^*/*IRF7^wt^*) and IRF7 overexpression (*CuO-IRF7*/*Control*) [35] at 0 h (mock) and 6 hpi with H6N2 infection (heatmap). Each cluster’s representative genes are indicated on the left of the heatmap, and biological process gene ontology terms and their p-values are indicated on the right.

**Figure 4 genes-11-00385-f004:**
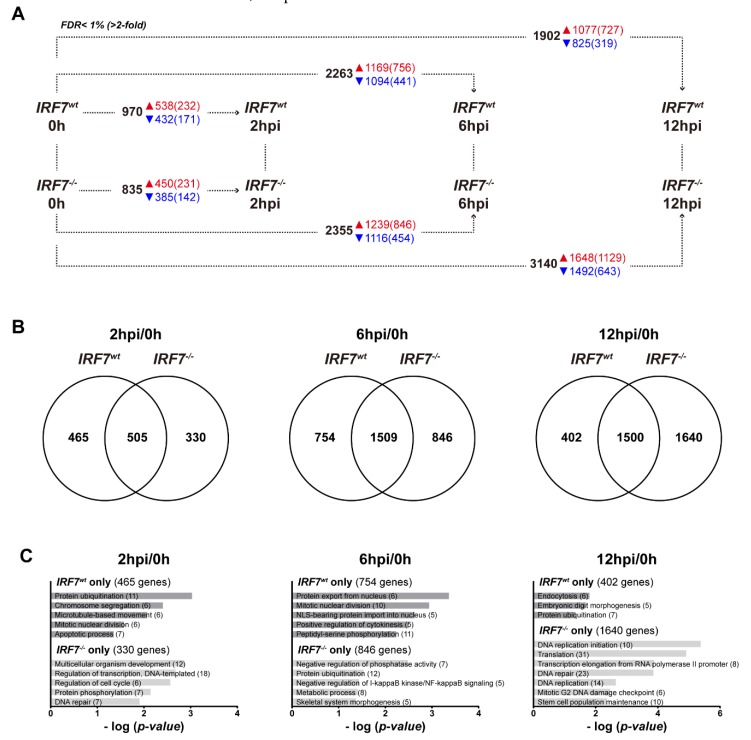
IRF7 deletion altered the host response against H6N2 infection (**A**) The number of differentially expressed genes (DEGs; FDR < 1%) for *IRF7^wt^* and *IRF7^-/-^* cell lines upon H6N2 infection at 2, 6 and 12 h post infections (hpi). Each horizontal comparison indicates the number of DEGs upon H6N2 infection at each time point compared to mock infection (0 h). The numbers of genes that have 2-fold expression changes are in the parentheses. (**B**) A Venn diagram of DEGs between *IRF7^wt^* and *IRF7^-/-^* cells upon H6N2 infection at each time point. (**C**) Gene ontology functional analysis was performed using each cell-specific DEG. The numbers of genes enriched in each biological process are in parentheses.

**Figure 5 genes-11-00385-f005:**
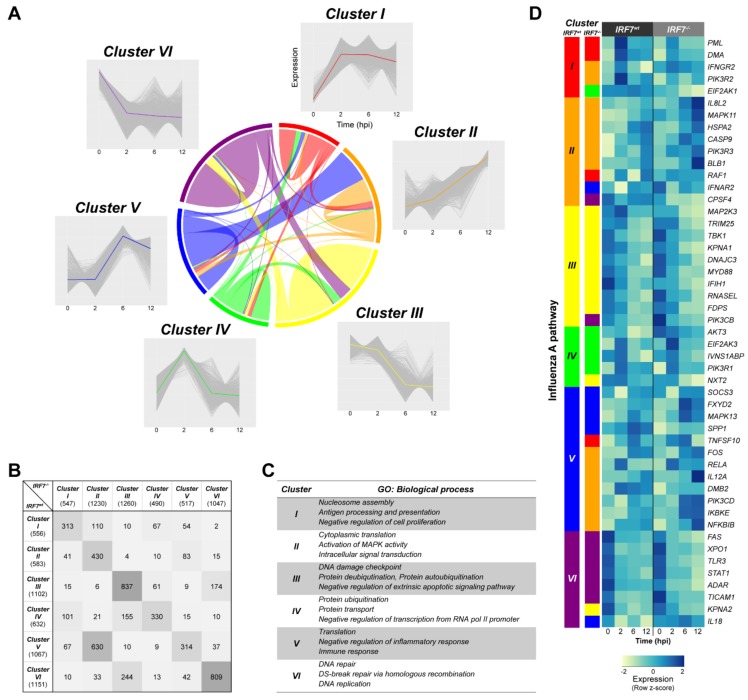
Gene expression profiles shifted as a result of IRF7 deletion in the host response. (**A**) The *k*-means clustering of differentially expressed genes (DEGs; *n* = 5091) and expression cluster shift of genes due to IRF7 knockout. Respective gene expression is shown in gray, and the representative expression pattern is highlighted in different colors. A chord diagram indicates the number of DEGs shifted clusters from *IRF7^wt^* to *IRF7^-/-^*. (**B**) The number of genes in each cluster. A complete list of genes in each cluster are in Appendix A. (**C**) The biological process gene ontology terms of *IRF7^wt^* DEGs in each cluster. (**D**) DEGs in the influenza A pathway and their expression clusters between *IRF7^wt^* and *IRF7^-/-^* are shown on the left. The colors of the clusters correspond to the colors used in (**A**). A heatmap of gene expression of the pathway is in the right panel. See Appendix A for each gene in the influenza A pathway.

**Figure 6 genes-11-00385-f006:**
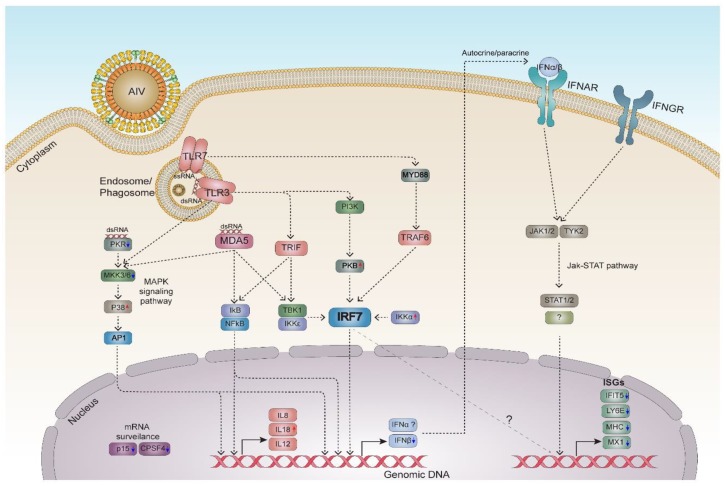
A proposed model of the molecular mechanism of chicken IRF7 modulation in the host response to avian influenza virus (AIV) infection A schematic model of the early antiviral response in chickens regarding IRF7 is proposed based on the DEGs in this study. AIV infection is sensed by pattern recognition receptors. TLR3 and TLR7 recognize endosomal/phagosomal double-stranded RNA (dsRNA) and single-stranded RNA (ssRNA) respectively. MDA5 and PKR respond to cytosolic dsRNA. This leads to a subsequent signaling cascade via adaptor molecules and protein kinases that activate the key transcription factors AP1, NF-κB and IRF7. The MAPK and PI3K/Akt pathways are potentially involved in the activation process. Activated IRF7 translocates to the nucleus and activates the transcription of IFNβ. AP1 and NF-κB also translocate to the nucleus and induce type I IFNs and inflammatory cytokines. Type I IFNs activate ISGs via the Jak–STAT pathway in an autocrine/paracrine manner through their receptors (IFNAR). IFIT5, interferon induced protein with tetratricopeptide repeats 5; JAK1/2, Janus kinase 1/2; LY6E, lymphocyte antigen 6 family member E; MKK3/6, MAPK kinase 3/6; MYD88, myeloid differentiation primary response 88; PKB, protein kinase B; STAT1/2, signal transducer and activator of transcription 1/2; TRAF6, TNF receptor associated factor 6; TRIF (TICAM1), toll like receptor adaptor molecule 1; TYK2, tyrosine kinase 2. Red and blue arrows indicate the up- and down-regulation of each gene respectively as a result of IRF7 knockout in the dataset. Further investigation is required for the items indicated by question marks.

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
