# Peer review of "Knockout of IRF7 Highlights its Modulator Function of Host Response Against Avian Influenza Virus and the Involvement of MAPK and TOR Signaling Pathways in Chicken"

_genes, 2020, doi:10.3390/genes11040385_

Round 1

Reviewer 1 Report

The authors addressed almost all the points made by the reviewer. 

I am still curious to get the opinion of the authors with respect to Figure 2b, where the Gene ontology (GO) and pathway analysis using DEGs from each time point, the number of genes enriched in each biological process are in parentheses. At two hours post infection by the KEGG pathway analysis it shows that six host genes related to herpesvirus simplex infection were enhances. Which are these genes? Does it menas that herpesvirus and avian influenza can enhance the expression of similar host genes. Usually herpesvirus infection. Although this is not important to the paper discussion,  however this reviewer wants to understand if this is an artifact of the analysis from which no inferences can be drawn?

Reviewer 2 Report

The authors answered all the questions raised by the reviewer. This reviewer has no further comment.

This manuscript is a resubmission of an earlier submission. The following is a list of the peer review reports and author responses from that submission.

Round 1

Reviewer 1 Report

In this manuscript the authors knock down the expression of the main transcription factor that mediates expression of type I interferon in chickens. The IRF7 expression was inactivated in the chicken embryonic fibroblast cell line DF-1 using the CRISPR/CAS9 system (IRF7-/-). The authors also introduced the SpCas9-puromycin resistant gene construct from the MCR plasmid to generate control (IRF7wt) DF-1 cells. Both cell lines were infected with H6N2 low pathogenic avian influenza virus. The level of virus replication on these cells was measured and showed that for IRF7-/- cells there was a significant increase on virus replication 12 hours post-inoculation.  Wide genomic RNA expression profiles of DF-1 IRF7-/- and DFIRF7wt together with integration of GO enrichment analysis plus gain and loss of function analysis using CuO-IRF7 cells with 2-fold IRF7 overexpression the authors concluded that similar to mammals, chicken IRF7 functions as a major modulator of the antiviral responses. Also, the authors concluded that in chickens the MAPK and mTOR signaling pathways may play important roles in modulating the type I IFN responses against avian influenza, and that other important mediators are working together with IRF7 to mount an antiviral response in chickens. Overall this manuscript is very well written and present genomic expression data that enhances the knowledge on how innate immune responses against avian influenza, in particular Type I antiviral responses, are regulated in chicken cells. Although is not this reviewer’s expertise, it was clear that a thorough analysis of the expression data was performed to untangle expression patterns that support that the expression of IFNB expression is controlled by IRF7. The authors need clarify the points listed below.

However, it appears that IRF7 other than type I IFN expression is involved in a wide range of biological processes (mitosis, cell cycle, protein modification, transcription and translation). The authors claim that despite the clear indication in the DEG lists and functional annotations, the expression level of individual genes did not show significant difference between the cell lines therefore they were not able to specify which functions are mainly regulated by IRF7.  Can the authors speculate how these potential functions of IRF7 can influence the innate immune response to AI? Please expand.

The authors claim that the shift observed from the stable-up-down pattern to constant up pattern once IRF7 is deleted suggests that the expression of alternative inflammatory response and immune response genes might be needed to compensate for the effect of deleted IRF7. From the discussion is not clear if this shift is unique to chicken cells (DF-1 cells). Has this shift unique to chicken cells? Has it been observed in mice epithelial cells? Please clarify.

Specifics

Line 18:  IRF7 knockout resulted in increased viral titers of low pathogenic avian influenza viruses. What influenza viruses? The authors only tested H6N2. Please clarify.

Line 33:  “However, current passive prevention practices such as quarantine and culling have been mainly applied to control AIV.” Is my understanding that there is not quarantine for flocks that are infected with low pathogenic influenza. Maybe in some instances there is vaccination but in the US culling is the main strategy to control further spread of the virus.

Line 195-196: Another strain of LPAIV H10N7 showed higher viral replication level as a result of IRF7 knockout 195 (data not shown). At what time points? Please clarify. If the data is not shown will it be better not to include it?

Figure 2b 2hpi. How can the authors explain the HSV-1 genes infection upregulated?

Lines 230 to 232: “Of note, the numbers of DEGs observed comparing IRF7wt with IRF7-/- cells were similar between up-regulated genes and down-regulated 231 genes at all time points” At all times post-infection?

Lines 274 to 276: “The number of DEGs did not have noticeable difference between IRF7wt and IRF7-/- cells at early time points but there were more than 1,200 DEGs (FDR < 1%) in IRF7-/- cells (3,140) than IRF7wt cells (1,902) 275 at 12hpi (Figure 4A).”  What exactly do the authors mean by noticeable differences. This sentence appears to show a contradiction.

Line 344: What MEF stands for?

Figure 6: This figure is very informative and helps to understand the pieces of the puzzle that have been emphasized with the interpretation of the results presented in this manuscript. However, the interpretation of upper (red) and lower (blue) arrows, the x symbol in the top of MDA5 and PKR boxes as well as the question mark symbols need to be included in the legend.

Reviewer 2 Report

In the present study, Kim and colleagues studied the function of IRF7 in regulating low pathogenic avian influenza virus replication in DF-1 cells. Using CRISPER/Cas9 system, they established the IRF7 knockout DF-1 cell lines. The low pathogenic H6N2 avian influenza virus showed better replication capacity in the IRF7 knockout cells. The transcriptome analysis of the low pathogenic avian influenza virus infected IRF7 knockout and wild type cells showed the association of IRF7 and MAPK and mTOR signaling pathway which may regulate virus replication.

  1. According to figure 2, the virus infection did not induce too much DEG between IRF7-/- and IRF7wt The MAPK signaling pathway was already changed even at 0 hpi (before virus infection). So it is very hard to conclude the change of MAPK signaling pathway is virus infection specific (related) or IRF7 modulates host response against avian influenza virus through MAPK pathway. Another thing is the pathways are enriched only by the significance of the DEG compared between IRF7-/- and IRF7wt cells, the change of most of the genes are probably less than 2-fold and the conclusion only based on RNA-seq data. At least, RT-PCR verification for some key molecules in MAPK signaling pathway should be included to support the authors’ conclusion.
  2. P9 line 270, for analysis “Differential expression analysis of host response upon H6N2 infection” and figure 4, for the shared DEG at different time points, do they include the DEG which showed different change direction (eg, upregulation in IRF7wt cells but downregulation in IRF7-/- cells or vice versa)? If yes, these DEG maybe very important for regulating virus replication and will be exclude in the GO analysis in figure 4C.
  3. P6 line 195, the authors claimed that the LPAIV H10N7 showed higher viral replication level in IRF7 knockout cells. Does that mean IRF7 specifically inhibit H6N2 LPAIV? Differential H6N2 replication was only observed at 12hpi and differential IFNB was only observed at 6hpi between IRF7 knockout and wild type cells. All these data challenge the function of IRF7 in regulating LPAIV replication. Do you observe the differential expression of ISG genes in the RNA-seq data?

Minor comments to the authors.

  1. P4 line 171, for figure S1C, what’s the templates used for PCR? Based on the primers showed in figure S1B (DEL-F/R), there should be a band for IRF7wtDF-1 and DF-1 (lane 1 and 3) but not lane 2 (IRF7wtDF-1+gRNA1&2) which co-transfected the gRNA for IRF7. Another thing is the authors did not show the detail size of the marker, but compare to the INT, the band in lane 2 (DEL) does not look like 2.7 kb.
  2. For figure 1D, why the y-axis of the viral titer was labeled as PFU+1/ml?
  3. P6 line 209, the expression of IFNB was not significant in IRF7-/- cells at 12hpi.
  4. P11 line 311, IL 18, IFNAR2, and IFNGR2 were not DEG shifted from cluster V to cluster II. Please double check.